# Use of psychological interventions among healthcare workers over the 2-year period following the COVID-19 pandemic: A longitudinal study

**Blanca García-Vázquez**[1,2], **Gonzalo Martínez-Alés**[3,4,5], **Eduardo Fernández-Jiménez**[1,5], **Jorge Andreo-Jover**[2,5], **Berta Moreno-Küstner**[6], **Sergio Minué**[7], **Fabiola Jaramillo**[7,8], **Inés Morán-Sánchez**[9], **Irene Martínez-Morata**[10], **José Luis Ayuso-Mateos**[2,4,11], **Carmen Bayón**[1,2,4,5], **María-Fe Bravo-Ortiz**[1,2,4,5], **Roberto Mediavilla**[2,4,11] *

1 Department of Psychiatry, Clinical Psychology, and Mental Health, Hospital Universitario La Paz, Madrid, Spain, 2 Department of Psychiatry, Universidad Autónoma de Madrid, Madrid, Spain, 3 CAUSALab, Harvard TH Chan School of Public Health, Harvard, Boston, Massachusetts, United States of America, 4 Centro de Investigación Biomédica en Red de Salud Mental (CIBERSAM), Instituto de Salud Carlos III, Madrid, Spain, 5 Hospital La Paz Institute for Health Research (IdiPAZ), Madrid, Spain, 6 Department of Personality, Assessment and Psychological Treatment, University of Málaga-IBIMA, Málaga, Spain, 7 Andalusian School of Public Health, Granada, Spain, 8 University of Chile, Santiago, Chile, 9 Murcia BioHealth Research Institute, University of Murcia, Murcia, Spain, 10 Department of Environmental Health Sciences, Columbia University Mailman School of Public Health, New York, New York, United States America, 11 Department of Psychiatry, La Princesa University Hospital - Instituto de Investigación Sanitaria Princesa (IIS-Princesa), Madrid, Spain

* roberto.mediavilla@uam.es

**Data Availability Statement:** All relevant data for this study are publicly available from the

## Abstract

### Introduction

Although healthcare workers (HCWs) have reported mental health problems since the beginning of the COVID-19 pandemic, they rarely use psychological support. Here, we described the use of psychological support among HCWs in Spain over the 2-year period following the initial pandemic outbreak and explore its association with workplace- and COVID-19-related factors measured at baseline, in 2020.

### Materials and methods

We conducted a longitudinal study on HCWs working in Spain. We used an online survey to collect information on sociodemographic characteristics, depressive symptoms, workplace- and COVID-19-related variables, and the use of psychological support at three time points (2020, 2021, and 2022). Data was available for 296, 294, and 251 respondents, respectively at time points 1, 2, and 3.

### Results

Participants had a median age of 43 years and were mostly females (n = 242, 82%). The percentage of HCWs using psychological support increased from 15% in 2020 to 23% in 2022. Roughly one in four HCWs who did not use psychological support reported symptoms

Repositorio de datos UAM (https://doi.org/10.21950/DX5TKS).

**Funding:** The study was funded by the Instituto de Salud Carlos III and the European Regional Development Fund (COV20/00988, awarded to M-FB-O). The work of RM was supported by the Instituto de Salud Carlos III and by the European Regional Development Fund (CD22/00061) and by the European Commission (grant agreement 101016127, awarded to J-LA-M). Funders were not involved in the study design; in the collection, analysis and interpretation of data; in the writing of the report; or in the deci-sion to submit the article for publication.

**Competing interests:** The authors have declared that no competing interests exist.

compatible with major depressive disorder at follow up. Baseline predictors of psychological support were having to make decisions about patients' prioritisation (OR 5.59, 95% CI 2.47, 12.63) and probable depression (wave 2: OR 1.12, 95% CI 1.06, 1.19; wave 3: OR 1.10, 95% CI 1.04, 1.16).

## Conclusions

Our results suggest that there is call for implementing mental health promotion and prevention strategies at the workplace, along with actions to reduce barriers for accessing psychological support.

## Introduction

Ever since the initial outbreak of the ongoing COVID-19 pandemic, research has consistently shown high rates of symptoms of anxiety, depression, or post-traumatic stress disorder (PTSD) among healthcare workers (HCWs) [1, 2]. In Spain, one of the earliest pandemic hot-spots worldwide, estimates showed that one in four HCWs had a probable major depressive disorder and one in five had a probable anxiety disorder or PTSD in 2020 [3, 4], and longitudinal analyses revealed that these problems either persist or get worse over time [5, 6]. This potentially threatens health systems with sick leaves due to mental health problems, including burnout.

Despite the prevalence and potential burden of mental health problems, HCWs rarely use mental health support. For instance, a representative study conducted in New York during the initial pandemic outbreak showed that only one in two HCWs seeking psychological support were actually receiving treatment [7]. In five Chinese regions including Hubei, only one in ten HCWs sought psychological support in February 2020 [8]. These findings suggest that, despite the early provision of mental health support to HCWs at the workplace during the pandemic [9], there are significant barriers to accessing psychological resources. Lack of trust in digital interventions was as a key barrier to psychological care in the general population, according to a recent umbrella review [10]. For HCWs, additional barriers included time constraints during the initial periods of the pandemic [11], internalized stigma [12], or the belief that HCWs never require psychological help [13]. In this study, we investigate the use of psychological support among HCWs in Spain over the two-year period following the initial outbreak of the pandemic and we explore the association between baseline workplace- and COVID-19-related factors and prospective use of psychological support.

## Methods

### Study design and participants

We conducted a longitudinal study including HCWs aged 18 and older working in healthcare facilities in Spain. We collected self-reported data via an online survey at three different time points (wave 1 from April 24th to June 22nd, 2020, n = 2,370 respondents; wave 2 from January 26th to March 8th, 2021, n = 1,807 respondents; and wave 3 form March 23rd to May 23rd, 2022, n = 538), which respectively concurred with the first pandemic outbreak, the third pandemic peak and the onset of vaccination campaigns among HCWs, and the fifth pandemic outbreak in Spain. We used non-probabilistic sampling techniques to recruit participants from outpatient or inpatient healthcare facilities, regardless of direct exposure to COVID-19 patients or involvement in clinical tasks (e.g., including cleaning staff). Data on the use of

psychological support was available for 296, 294, and 251 respondents, respectively, at waves 1, 2, and 3. These respondents are the sample of the present study. Because recruitment was opened to new participants at each time point, we have a total of 328 unique respondents, of whom 204 (62%) completed the three assessment waves, 105 (32%) completed two, and 19 (6%) completed one. Data on psychological support were missing for a large proportion of the study sample because the survey item on psychological support pertained in the last section of the survey—we assumed data missingness to be completely at random. A detailed baseline description of the full sample is available elsewhere [4]. Authors had access to personal data during the curation process and the dataset could not be fully anonymised in accordance with the General Data Protection Regulation (GDPR) of the European Union.

## Measures and procedures

At baseline, we collected information on age, gender (binary), and depression symptoms, as measured by the Spanish version of the 9-item Patient Health Questionnaire (PHQ-9) [14, 15]. The PHQ-9 uses a Likert scale ranging from 0 ("not at all") to 3 ("nearly every day") to explore diagnostic criteria of major depressive disorder according to the fourth version of the Diagnostic and Statistical Manual of Mental Disorders (DSM-IV). Items cover apathy, anhedonia, hopelessness, or sleeping problems, among other symptoms. The PHQ-9 total score ranges from 0 to 27 and a cut-off score of 10 points suggests probable major depressive disorder. As part of the baseline procedures of the current study, we reported a Cronbach's alpha of 0.88 (95 percent CI from 0.87 to 0.89) [4]. We also collected information on the following workplace- and COVID-19-related variables: type of job, direct exposure to COVID-19 patients, adequate access to personal protective equipment, social stigma for working with COVID-19 patients, decision making on patient prioritization, and perceived social network support at work. Type of job included four categories, namely physicians (e.g., medical doctors, clinical psychologists), nurses, ancillary workers (including nursing and other health technicians), and others. To analyse direct exposure to COVID-19 patients, we asked whether the person had been close to patients who were suspected or confirmed cases of COVID-19 in the previous two weeks (yes/no/not sure); to explore adequate access to protective equipment, we asked it the person considered it enough to avoid getting the virus? (response options ranged from "completely insufficient" to "sufficient"); to explore stigma and discrimination, we asked whether the person had felt stigmatized or discriminated against due to being a HCW (from "strongly disagree" to "strongly agree"); to explore decision making, we asked whether the person had to prioritise patients with COVID-19 to, for example, receive intensive treatment and/ or mechanical ventilation (yes/no/does not apply; last, social network at work was directly assessed with the item "I have a reliable network of supportive colleagues at work", with response options from "strongly disagree" to "strongly agree". Ordinal variables were dichotomised to ease the interpretability of the results. Information on the use of psychological support in the previous three months (yes/no) was collected at all timepoints.

## Statistical analyses

First, we reported categorical variables as frequencies and percentages and continuous variables as mean and standard deviations. We stratified them by whether the person used psychological support and presented them across time points. Then, we used binary logistic regression models to explore the association between (i) baseline measures of probable major depressive disorder and of workplace- and COVID-19-related variables and (ii) use of psychological support, measured at every wave. We provide odds ratios with 95 percent confidence intervals. All analyses were run using R and R Studio.

## Ethical considerations

The study followed the World Medical Association's Declaration of Helsinki. The protocol obtained ethics approval from the Ethics Committee at Hospital Universitario La Paz in Madrid (identifier PI- 4099) and was ratified by all participating sites. All participants gave written informed consent before taking part in the trial.

## Results and discussion

Table 1 summarizes the characteristics of the sample at the three time points. At baseline, participants had a median age of 43 years, and most were females (n = 242, 82%) and worked as nurses (n = 97, 33%) or physicians (n = 108, 37%). More than half had been exposed to COVID-19 patients (n = 154, 62%) and almost one in three scored above the threshold for probable depression on the PHQ-9 (n = 88, 30%).

The number and proportion of HCWs who had received psychological support during the previous three months increased over time from n = 44 (15%) at baseline to n = 51 (18%) in survey wave 2 and n = 59 (23%) in survey wave 3. Overall, workplace- and COVID-19-related variables at baseline were not associated with use of psychological support at follow-up (see Table 2): decision making on patient prioritization at baseline was associated with having received psychological support at survey wave 2 (OR 5.6, 95 percent CI 2.5, 12.6) and probable major depressive disorder at baseline was associated with psychological support at waves 2 (OR 1.1, 95 percent CI 1.1, 1.2) and 3 (OR 1.1, 95 percent CI 1.0, 1.2).

**Table 1. Characteristics of the participants.**

| | Wave 1 | | | Wave 2 | | | Wave 3 | | |
|---|---|---|---|---|---|---|---|---|---|
| | Overall | Not received | Received | Overall | Not received | Received | Overall | Not received | Received |
| | (N = 296) | (n = 252, 85%) | (n = 44, 15%) | (N = 294) | (n = 243, 83%) | (n = 51, 17%) | (N = 251) | (n = 192, 76%) | (n = 59, 14%) |
| Age in years, Median (IQR) | 43 (11) | 43 (11) | 40 (11) | 43 (11) | 44 (11) | 37 (10) | 43 (11) | 44 (11) | 38 (10) |
| Gender [female], n (%) | 242 (82%) | 202 (80%) | 40 (91%) | 238 (83%) | 191 (81%) | 47 (92%) | 200 (81%) | 148 (78%) | 52 (91%) |
| Type of job | | | | | | | | | |
| Physician | 108 (37%) | 92 (37%) | 16 (36%) | 103 (37%) | 86 (37%) | 17 (36%) | 82 (34%) | 64 (35%) | 18 (34%) |
| Nurse | 97 (33%) | 78 (31%) | 19 (43%) | 92 (33%) | 70 (30%) | 22 (47%) | 82 (34%) | 56 (30%) | 26 (49%) |
| Ancillary workers | 36 (12%) | 32 (13%) | 4 (9.1%) | 28 (10%) | 24 (10%) | 4 (8.5%) | 27 (11%) | 22 (12%) | 5 (9.4%) |
| Other | 53 (18%) | 48 (19%) | 5 (11%) | 54 (19%) | 50 (22%) | 4 (8.5%) | 47 (20%) | 43 (23%) | 4 (7.5%) |
| Direct exposure to COVID-19 patients [yes] [¥] | 154 (62%) | 126 (61%) | 28 (68%) | 143 (61%) | 114 (59%) | 29 (74%) | 122 (61%) | 93 (61%) | 29 (62%) |
| Access to protective equipment [inadequate] [¥] | 159 (55%) | 133 (54%) | 26 (59%) | 145 (54%) | 115 (52%) | 30 (64%) | 122 (52%) | 97 (53%) | 25 (47%) |
| Social stigma for working with COVID-19 patients [yes] [¥] | 91 (31%) | 73 (29%) | 18 (41%) | 85 (31%) | 65 (28%) | 20 (43%) | 71 (30%) | 53 (29%) | 18 (33%) |
| Patient prioritization [yes] | 48 (25%) | 34 (21%) | 14 (47%) | 40 (23%) | 23 (16%) | 17 (52%) | 38 (25%) | 26 (23%) | 12 (31%) |
| Social support from colleagues [yes] [¥] | 265 (90%) | 226 (90%) | 39 (89%) | 238 (90%) | 194 (89%) | 44 (98%) | 201 (89%) | 155 (88%) | 46 (90%) |
| Probable major depressive disorder (PHQ-9 > 9) [yes] | 88 (30%) | 69 (28%) | 19 (43%) | 72 (27%) | 51 (24%) | 21 (46%) | 59 (26%) | 41 (23%) | 18 (35%) |

Note.

All percentages are valid percentages. PHQ-9 = Patients' Health Questionnaire—9 items.

[¥] Ordinal variable dichotomized to facilitate the interpretation of the table

**Table 2. Association between workplace- and COVID-19-related variables at baseline and use of psychological support at follow-up.**

|  | Wave 2 | | Wave 3 | |
|---|---|---|---|---|
|  | **Odds ratio** | **95% CI** | **Odds ratio** | **95% CI** |
| Direct exposure to COVID-19 patients [yes] | 2.03 | (0.93, 4.41) | 1.03 | (0.53, 2.03) |
| Access to protective equipment [yes] | 0.81 | (0.55, 1.19) | 1.10 | (0.75, 1.60) |
| Social stigma for working with COVID-19 patients [yes] | 1.20 | (0.84, 1.71) | 1.13 | (0.80, 1.60) |
| Patient prioritization [yes] | 5.59 | (2.47, 12.63) | 1.45 | (0.64, 3.26) |
| Social support from colleagues [yes] | 1.45 | (0.88, 2.41) | 1.17 | (0.75, 1.85) |
| Probable major depressive disorder (PHQ-9 > 9) [yes] | 1.12 | (1.06, 1.19) | 1.10 | (1.04, 1.16) |

Note.

PHQ-9 = Patients' Health Questionnaire—9 items.

Roughly one in four HCWs who did not use psychological support reported symptoms compatible with major depressive disorder at follow up, suggesting barriers in access to psychological support among HCWs in need for it. Importantly, we found that the proportion of HCWs who received psychological support increased over time, consistent with the hypothesis that increased workload during the initial pandemic outbreak was a driver of barriers in access to psychological support [16]. In addition, this is the first study to examine the association between baseline workplace- and COVID-19-related variables and prospective use of psychological support among HCWs: Use of psychological support was overall not associated with job type, level of exposure to COVID-19 patients, or fear of infection—in spite of their link to worse mental health [3, 17]. HCWs involved in patient prioritization during the initial pandemic outbreak were more likely to receive psychological support one year later. While this result is not easy to interpret, a potential causal relationship between involvement in patient prioritization (especially in Spring 2020, when healthcare systems could not cope with increases in acute- and critical-care requirements), negative mental health outcomes, and subsequent need for psychological support seems plausible.

Our study has some limitations. First, we used non-probabilistic sampling techniques during recruitment, which increases the risk of self-selection bias. It is thus plausible that our sample is composed of HCWs who are likely to report mental health distress, and therefore underrepresents those without distress or not likely to report it. Second, because we were interested in longitudinal and prospective associations, we only included respondents to follow-up assessments. This may increase self-selection bias. Third, we did not collect any information on the specific type of psychological support, which would have been helpful to provide specific recommendations depending on the intervention providers, settings, or duration. Last, primary outcomes are self-reported, which may introduce additional bias (e.g., recall bias).

Our findings have implications for the design and implementation of mental health actions for HCWs in crisis settings which are in line with guidelines from the World Health Organization and the International Labour Organization [18]. First, we call for a generalised reduction of psychosocial risks at work that extend beyond the midst of initial pandemic crises, as the use of psychological support in 2021 and 2022 was at least as frequent as it was in 2020. Second, health promotion strategies at work should target barriers for accessing to psychological support among HCWs, which include not only contextual factors such as rotating shifts or work-life balance, but also internal factors, such as internalised stigma caused by seeking or using psychological support [12], or fear for the negative consequences of a mental health diagnosis [13]. Last, we found only a 10% increase of use of psychological support among people with probable major depressive disorder, compared with people without it. In light of this, we call

for improving access to evidence-based scalable interventions at the workplace (see for example [19]).

## Conclusions

Our results suggest that many HCWs may use psychological support in the 2-year period following the initial pandemic outbreak regardless of prior exposure to workplace- and COVID-19-related stressors. Specific mental health actions should focus on health promotion and primary prevention as well as on supporting people with mental health problems, such as depression symptoms.

## Supporting information

**S1 Checklist. STROBE checklist.** STROBE checklist for observational studies.
(DOCX)

## Author Contributions

**Conceptualization:** Blanca García-Vázquez, Gonzalo Martínez-Alés, Eduardo Fernández-Jiménez, Berta Moreno-Küstner, Sergio Minué, Fabiola Jaramillo, Inés Morán-Sánchez, Irene Martínez-Morata, María-Fe Bravo-Ortiz, Roberto Mediavilla.

**Data curation:** Roberto Mediavilla.

**Formal analysis:** Roberto Mediavilla.

**Funding acquisition:** José Luis Ayuso-Mateos, María-Fe Bravo-Ortiz, Roberto Mediavilla.

**Investigation:** Jorge Andreo-Jover, Berta Moreno-Küstner, Sergio Minué, Fabiola Jaramillo, Inés Morán-Sánchez, Irene Martínez-Morata.

**Methodology:** Blanca García-Vázquez, Gonzalo Martínez-Alés, Roberto Mediavilla.

**Project administration:** Roberto Mediavilla.

**Software:** Roberto Mediavilla.

**Supervision:** José Luis Ayuso-Mateos, Carmen Bayón, María-Fe Bravo-Ortiz, Roberto Mediavilla.

**Validation:** Gonzalo Martínez-Alés, Roberto Mediavilla.

**Visualization:** Gonzalo Martínez-Alés, Roberto Mediavilla.

**Writing – original draft:** Blanca García-Vázquez, Roberto Mediavilla.

**Writing – review & editing:** Gonzalo Martínez-Alés, Eduardo Fernández-Jiménez, Jorge Andreo-Jover, Berta Moreno-Küstner, Sergio Minué, Fabiola Jaramillo, Inés Morán-Sánchez, Irene Martínez-Morata, Carmen Bayón.

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
