## [Decision Letter · Decision Letter 0]

20 Jul 2023

PONE-D-23-19134Use of psychological interventions among healthcare workers over the 2-year period following the covid-19 pandemic: A prospective cohort studyPLOS ONE

Dear Dr. Mediavilla,

Thank you for submitting your manuscript to PLOS ONE. After careful consideration, we feel that it has merit but does not fully meet PLOS ONE’s publication criteria as it currently stands. Therefore, we invite you to submit a revised version of the manuscript that addresses the points raised during the review process.

One of our associate editors and two reviewers carefully read the manuscript. Based on their evaluations the manuscript is Major Revision. The associate editor provided the following reasons:

The manuscript needs to be rewritten, taking into account the following main comments made by the reviewers

Methods: more details are needed

Please revise the manuscript according to the reviewers' comments and upload the revised file. Any revisions should be “clearly highlighted”, for example using the Track Changes function in Microsoft Word or using a different colour, so that they are easily visible to the editors and reviewers. Please provide a short Cover letter detailing any changes, for the benefit of the editors and reviewers.

We look forward to receiving your revised manuscript.

Kind regards,

Juan Jesús García-Iglesias, Ph.D.

Academic Editor

PLOS ONE

Journal Requirements:

Reviewers' comments:

Reviewer's Responses to Questions

**Comments to the Author**

1. Is the manuscript technically sound, and do the data support the conclusions?

Reviewer #1: Yes

Reviewer #2: No

2. Has the statistical analysis been performed appropriately and rigorously? 

Reviewer #1: Yes

Reviewer #2: No

3. Have the authors made all data underlying the findings in their manuscript fully available?

Reviewer #1: Yes

Reviewer #2: No

4. Is the manuscript presented in an intelligible fashion and written in standard English?

Reviewer #1: Yes

Reviewer #2: Yes

5. Review Comments to the Author

Reviewer #1: The present manuscript presents the use of psychological support among health care workers in Spain over t 2 years period and the predictors. It is a quite short report but well-written. I have some points of suggestions for further increase the quality of the paper

1. Introduction: Would be better if you could provide some data about the burden of HCWs in Spain during the COVID-19 pandemic for better understanding of the importance of this study. Also, if any, any previous findings about mental health issues of HCWs around the world, especially in Spain.

2. Methods-Measures and Procedures: Please describe a little bit more about the PHQ-9 as the instrument being used: e.g., some items, response options, etc.

3. Methods-Measures and Procedures: Describe in more details the questions and response options for workplace- and COVID-19-related questions to give the readers better understanding of the survey

4. Results: Do you have any further information about the type, duration, and provider of the psychological support that they received? Online or offline, etc? This might be very interesting to know

5. Table 1: for the number of overall, not received and received in all three waves, please write in n (%) too, just as the other results.

6. Table 2: Give sign that shows statistically significant results

7. Table 2: You analysed workplace- and COVID-19-related variables at baseline associations with use of psychological support at follow-up? How about the use of psychological support in wave 3 based on wave 2? Or you only collect the workplace- and COVID-19-related variables at baseline but not at other waves?

8. Page 8, lines 14-18: Your second suggestion is quite contrary to your findings: this study found that work-related and stigma were not associated with the use of psychological support. Please revise.

Reviewer #2: Thanks for the opportunity to review this manuscript. I hope you can adress the points below:

1. Summary of the research and overall impression

The study investigates the relationship between help-seeking/use of psychological support and workplace and COVID-19-related factors, such as social support from colleagues and direct exposure to COVID-19 patients.

Overall, the methods section needs to be clarified for the reader to understand what exactly has been done and how the authors reached their conclusion. In addition, tables need to be self-explanatory, as some readers check the tables before reading the entire manuscript.

Below, you can find some comments on the manuscript with examples:

2. Evidence and examples.

Major Issues

1. The authors claim to have conducted a cohort study, but from the methods described, this appears to be a cross-sectional study with data collected at three different time points. For this study to be a cohort study, the same participants should be followed over time to compare the outcome of interest in both exposed and not exposed groups.

2. Methods, especially data collection procedure, need to be clarified to avoid confusion. For example, on page 3, line 22, authors said, “wave 1 from April 24th to June 22nd, 2020, n = 2,370 respondents; wave 2 from January 26th 22 to March 8th, 2021, n = 1,807 respondents; and wave 3 form March 23rd to May 23rd 23 , 2022, n = 538)”. Then on page 4, line 5, say said, “Data on the use of psychological support was available for 296, 294, and 251 respondents, respectively, at waves 1, 2, and 3. Two thirds of participants (n = 204, 62%) filled in the three assessment waves, one third (n = 105, 32%) filled in two assessment waves, and 19 (6%) completed one wave only”. I’m not sure which assessment is referred to in the last sentences and who the numbers are related to each other.

3. Page 4, line 20, the authors said, “At each follow up assessment, we collected information on use of psychological support in 21 the previous three months (yes/no)”. Was it only collected at the follow up assessment, as it is also mentioned in the baseline (supposing that wave 1 is the baseline)?

4. What is the association between having depressive symptoms and seeking help/receiving psychological support? It is an important relationship that should be highlighted.

5. In Table 1, the percentages are not consistent across the table, and it is not clear what they refer to, which makes it confusing to the reader.

6. PLOS authors have the option to publish the peer review history of their article (what does this mean?). If published, this will include your full peer review and any attached files.

Reviewer #1: No

Reviewer #2: No

---

## [Author Response · Author response to Decision Letter 0]

21 Aug 2023

[Please use the pdf version of this response letter to make sure that it keeps its original format]

Dear editor and reviewers, 

Thank you very much for your revision on the manuscript entitled “Use of psychological interventions among healthcare workers over the 2-year period following the COVID-19 pandemic: A prospective cohort study”, which I submitted to PLOS ONE on behalf of my co-authors. 

I have addressed your comments one by one. You will find them below in bold, followed by our replies and the modified sections of the manuscript (in blue). Please see the attached pdf file, which keeps the correct format of this letter. I have also uploaded two copies of the manuscript to the system, with and without tracked changes. 

I very much appreciate your suggestions, which I think have improved the manuscript substantially. 

Best wishes, 

The corresponding author 

Editor

I have updated both the title page and manuscript to comply with journal requirements.

I have removed the section entitled “Role of the funding source” from the manuscript.

We used pseudonymized data to support the conclusions of the current study. Our dataset involves human participants, many of whom have active medical records. The dataset contains both sociodemographic information (e.g., age, gender, type of job) and specific work-related data (e.g., having to make decisions on patients’ prioritization) which critically hinder the anonymization process, according to the General Data Protection Regulation (GDPR) of the European Union. 

Our data availability statement now reads:

The supporting dataset contains data that can identify individual participants and cannot be fully anonymized in accordance with the General Data Protection Regulation (GDPR) of the European Union.

We also modified the end of the “Study design and participants” section:

Authors had access to personal data during the curation process and the dataset could not be fully anonymised in accordance with the General Data Protection Regulation (GDPR) of the European Union. 

I have moved the Ethics statement and placed it at the end of the Methods section.

 

Reviewer #1

The present manuscript presents the use of psychological support among health care workers in Spain over t 2 years period and the predictors. It is a quite short report but well-written. I have some points of suggestions for further increase the quality of the paper.

Thank you very much for your revision. We have expanded the manuscript to provide more detail, following your suggestions, especially regarding the Methods section. We think it is now more replicable but still accessible.

1. Introduction: Would be better if you could provide some data about the burden of HCWs in Spain during the COVID-19 pandemic for better understanding of the importance of this study. Also, if any, any previous findings about mental health issues of HCWs around the world, especially in Spain.

Thank you very much for this. More context was indeed required to help the reader understand the practical implications of our findings. We have expanded on the Spanish context because we know it quite well and because our findings may not have the same implications for countries with different health systems -e.g., the US. We used two cohort studies that used longitudinal data. 

The text now reads:

In Spain, one of the earliest pandemic hotspots worldwide, estimates showed that one in four HCWs had a probable major depressive disorder and one in five had a probable anxiety disorder or PTSD in 2020 (3,4), and longitudinal analyses revealed that these problems either persist or get worse over time (5,6). This potentially threatens health systems with sick leaves due to mental health problems, including burnout.

2. Methods-Measures and Procedures: Please describe a little bit more about the PHQ-9 as the instrument being used: e.g., some items, response options, etc.

We now describe the PHQ-9 more in depth. We also provide Cronbach’s alpha for the baseline sample of the current study, published in a previous report. Following Reviewer #2, we also did a new sensitivity analysis to explore the association between depression symptoms and use of psychological support. 

The section now reads:

At baseline, we collected information on age, gender (binary), and depression symptoms, as measured by the Spanish version of the 9-item Patient Health Questionnaire (PHQ-9) (14,15). The PHQ-9 uses a Likert scale ranging from 0 (“not at all”) to 3 (“nearly every day”) to explore diagnostic criteria of major depressive disorder according to the fourth version of the Diagnostic and Statistical Manual of Mental Disorders (DSM-IV). Items cover apathy, anhedonia, hopelessness, or sleeping problems, among other symptoms. The PHQ-9 total score ranges from 0 to 27 and a cut-off score of 10 points suggests probable major depressive disorder. As part of the baseline procedures of the current study, we reported a Cronbach’s alpha of 0.88 (95 percent CI from 0.87 to 0.89) (4).

3. Methods-Measures and Procedures: Describe in more details the questions and response options for workplace- and COVID-19-related questions to give the readers better understanding of the survey

Thank you very much for your comment. We have expanded the section describing the baseline measures and procedures. It now reads:

At baseline, we collected information on age, gender (binary), and depression symptoms, as measured by the Spanish version of the 9-item Patient Health Questionnaire (PHQ-9) (14,15). The PHQ-9 uses a Likert scale ranging from 0 (“not at all”) to 3 (“nearly every day”) to explore diagnostic criteria of major depressive disorder according to the fourth version of the Diagnostic and Statistical Manual of Mental Disorders (DSM-IV). Items cover apathy, anhedonia, hopelessness, or sleeping problems, among other symptoms. The PHQ-9 total score ranges from 0 to 27 and a cut-off score of 10 points suggests probable major depressive disorder. As part of the baseline procedures of the current study, we reported a Cronbach’s alpha of 0.88 (95 percent CI from 0.87 to 0.89) (4). We also collected information on the following workplace- and COVID-19-related variables: type of job, direct exposure to COVID-19 patients, adequate access to personal protective equipment, social stigma for working with COVID-19 patients, decision making on patient prioritization, and perceived social network support at work. Type of job included four categories, namely physicians (e.g., medical doctors, clinical psychologists), nurses, ancillary workers (including nursing and other health technicians), and others. To analyse direct exposure to COVID-19 patients, we asked whether the person had been close to patients who were suspected or confirmed cases of COVID-19 in the previous two weeks (yes/no/not sure); to explore adequate access to protective equipment, we asked it the person considered it enough to avoid getting the virus? (response options ranged from “completely insufficient” to “sufficient”); to explore stigma and discrimination, we asked whether the person had felt stigmatized or discriminated against due to being a HCW (from “strongly disagree” to “strongly agree”); to explore decision making, we asked whether the person had to prioritise patients with COVID-19 to, for example, receive intensive treatment and/or mechanical ventilation (yes/no/does not apply; last, social network at work was directly assessed with the item “I have a reliable network of supportive colleagues at work”, with response options from “strongly disagree” to “strongly agree”. Ordinal variables were dichotomised to ease the interpretability of the results. Information on the use of psychological support in the previous three months (yes/no) was collected at all timepoints. 

4. Results: Do you have any further information about the type, duration, and provider of the psychological support that they received? Online or offline, etc? This might be very interesting to know

Thank you for your question. We did not collect any details about the psychological support that people received. We agree that this would be very useful to inform policy makers now, but it was unfortunately not feasible to extend the survey. We have included a sentence in the Limitations section to acknowledge this:

Our study has some limitations. First, we used non-probabilistic sampling techniques during recruitment, which increases the risk of self-selection bias. It is thus plausible that our sample is composed of HCWs who are likely to report mental health distress, and therefore underrepresents those without distress or not likely to report it. Second, because we were interested in longitudinal and prospective associations, we only included respondents to follow-up assessments. This may increase self-selection bias. Third, we did not collect any information on the specific type of psychological support, which would have been helpful to provide specific recommendations depending on the intervention providers, settings, or duration. Last, primary outcomes are self-reported, which may introduce additional bias (e.g., recall bias).

5. Table 1: for the number of overall, not received and received in all three waves, please write in n (%) too, just as the other results.

Thank you very much. We described this in the Results section because we found it a bit confusing to mix both column and row percentages in Table 1. We have however reconsidered it based on your comment and we have updated the table accordingly. We think it is now clear.

Table 1. Characteristics of the participants

 Wave 1 Wave 2 Wave 3

 Overall Not received Received Overall Not received Received Overall Not received Received

 (N = 296) (n = 252, 85%) (n = 44, 15%) (N = 294) (n = 243, 83%) (n = 51, 17%) (N = 251) (n = 192, 76%) (n = 59, 14%)

Age in years, Median (IQR) 43 (11) 43 (11) 40 (11) 43 (11) 44 (11) 37 (10) 43 (11) 44 (11) 38 (10)

Gender [female], n (%) 242 (82%) 202 (80%) 40 (91%) 238 (83%) 191 (81%) 47 (92%) 200 (81%) 148 (78%) 52 (91%)

Type of job 

Physician 108 (37%) 92 (37%) 16 (36%) 103 (37%) 86 (37%) 17 (36%) 82 (34%) 64 (35%) 18 (34%)

Nurse 97 (33%) 78 (31%) 19 (43%) 92 (33%) 70 (30%) 22 (47%) 82 (34%) 56 (30%) 26 (49%)

Ancillary workers 36 (12%) 32 (13%) 4 (9.1%) 28 (10%) 24 (10%) 4 (8.5%) 27 (11%) 22 (12%) 5 (9.4%)

Other 53 (18%) 48 (19%) 5 (11%) 54 (19%) 50 (22%) 4 (8.5%) 47 (20%) 43 (23%) 4 (7.5%)

Direct exposure to COVID-19 patients [yes] ¥ 154 (62%) 126 (61%) 28 (68%) 143 (61%) 114 (59%) 29 (74%) 122 (61%) 93 (61%) 29 (62%)

Access to protective equipment [inadequate] ¥ 159 (55%) 133 (54%) 26 (59%) 145 (54%) 115 (52%) 30 (64%) 122 (52%) 97 (53%) 25 (47%)

Social stigma for working with COVID-19 patients [yes] ¥ 91 (31%) 73 (29%) 18 (41%) 85 (31%) 65 (28%) 20 (43%) 71 (30%) 53 (29%) 18 (33%)

Patient prioritization [yes] 48 (25%) 34 (21%) 14 (47%) 40 (23%) 23 (16%) 17 (52%) 38 (25%) 26 (23%) 12 (31%)

Social support from colleagues [yes] ¥ 265 (90%) 226 (90%) 39 (89%) 238 (90%) 194 (89%) 44 (98%) 201 (89%) 155 (88%) 46 (90%)

Probable major depressive disorder (PHQ-9 > 9) [yes] 88 (30%) 69 (28%) 19 (43%) 72 (27%) 51 (24%) 21 (46%) 59 (26%) 41 (23%) 18 (35%)

Note. 

All percentages are valid percentages. PHQ-9 = Patients' Health Questionnaire - 9 items. 

¥ Ordinal variable dichotomized to facilitate the interpretation of the table

6. Table 2: Give sign that shows statistically significant results

Thank you very much. Following the recommendations of the International Committee of Medical Journal Editors (ICMJE), we did not report p values because they “fail to convey important information about effect size and precision of estimates” (https://www.google.com/url?sa=t&rct=j&q=&esrc=s&source=web&cd=&ved=2ahUKEwihgZjto-2AAxXEZaQEHXovBtoQFnoECBUQAQ&url=https%3A%2F%2Fwww.icmje.org%2Ficmje-recommendations.pdf&usg=AOvVaw2fN9QyBkH-854oixzS9fO8&opi=89978449, page 17). Considering that PlosONE submission guidelines do not specifically ask to provide this information, we decided to report 95 percent confidence intervals, which offer a more detail description of the precision of the estimate and can also be used for hypothesis testing –please note that the Abstract and the Results section only highlights the confidence interval that does not include a value of zero: 

Abstract:

Baseline predictors of psychological support were having to make decisions about patients’ prioritisation (OR 5.59, 95% CI 2.47, 12.63) and probable depression (wave 2: OR 1.12, 95% CI 1.06, 1.19; wave 3: OR 1.10, 95% CI 1.04, 1.16).

Results:

Overall, workplace- and COVID-19-related variables at baseline were not associated with use of psychological support at follow-up (see Table 2): decision making on patient prioritization at baseline was associated with having received psychological support at survey wave 2 (OR 5.6, 95 percent CI 2.5, 12.6) and probable major depressive disorder at baseline was associated with psychological support at waves 2 (OR 1.1, 95 percent CI 1.1, 1.2) and 3 (OR 1.1, 95 percent CI 1.0, 1.2).

7. Table 2: You analysed workplace- and COVID-19-related variables at baseline associations with use of psychological support at follow-up? How about the use of psychological support in wave 3 based on wave 2? Or you only collect the workplace- and COVID-19-related variables at baseline but not at other waves?

Thank you for raising this very interesting question. We are indeed interested in the association between workplace- and COVID-19-related variables at baseline and use of psychological support at one and two years, because our aim was to explore the mid- and long-term probable effects of exposure to stressors on mental health. As you noticed, most of these variables were only collected at baseline because we tried to capture a very extraordinary and complex reality which was fortunately over in 2021 –e.g., shortages in protective equipment, forced prioritization of patients in intensive care units, or violence towards frontline HCWs because they could infect other people.

8. Page 8, lines 14-18: Your second suggestion is quite contrary to your findings: this study found that work-related and stigma were not associated with the use of psychological support. Please revise.

Thank you very much for noticing this, which indeed requires some clarification. In our study, we measured self-reported stigma and discrimination due to being a HCW. Our rationale for this were reports in Spain and other countries on violence towards HCWs (e.g., in Madrid, there were cases of neighbours asking nurses and doctors to leave their apartments and sleep in hotels during the first pandemic outbreak). In the Discussion, we refer to another type of stigma, namely the internalized stigma caused by seeking or using psychological support. We now that this is an important barrier to the uptake of mental health services, especially among HCWs, and so we call for targeted actions within psychological support programmes. 

The Methods section now clarifies that we refer to stigma due to being a HCW and reads as follows:

to explore stigma and discrimination, we asked whether the person had felt stigmatized or discriminated against due to being a HCW (from “strongly disagree” to “strongly agree”)

In the Discussion, we also clarified that we refer to stigma due to receiving psychological support:

Second, health promotion strategies at work should target barriers for accessing to psychological support among HCWs, which include not only contextual factors such as rotating shifts or work-life balance, but also internal factors, such as internalised stigma caused by seeking or using psychological support (12), or fear for the negative consequences of a mental health diagnosis (13).

 

Reviewer 2

Thanks for the opportunity to review this manuscript. I hope you can address the points below:

Thank you very much for your comments. We addressed them one by one (see below).

The study investigates the relationship between help-seeking/use of psychological support and workplace and COVID-19-related factors, such as social support from colleagues and direct exposure to COVID-19 patients. Overall, the methods section needs to be clarified for the reader to understand what exactly has been done and how the authors reached their conclusion. In addition, tables need to be self-explanatory, as some readers check the tables before reading the entire manuscript.

Thank you very much. We have extended the manuscript to provide a more detailed description of the Methods we used in this study. You can find them highlighted in blue in the tracked manuscript and in response to your specific questions below. We have also uploaded the STROBE checklist as a supporting file to facilitate the reading and the revision process. We have also included more information in both tables (see below)

1. The authors claim to have conducted a cohort study, but from the methods described, this appears to be a cross-sectional study with data collected at three different time points. For this study to be a cohort study, the same participants should be followed over time to compare the outcome of interest in both exposed and not exposed groups.

Thank you for your comment. We recruited a cohort of HCWs in 2020 (baseline) and invited them to complete the survey again in 2021(t2) and 2022 (t3). Many participants were lost to follow up during the study and others were invited to participate in t2 and t3, regardless of whether they were already part of the study (some authors call this an open cohort study). Nonetheless, we understand your concern, as the term “cohort” may be misunderstood by some readers who expect a closed group of people followed up over a certain period (this is also related to your next comment). We have therefore removed the term “cohort” from the manuscript and use only the descriptive “prospective” to inform the reader about the design of the study (longitudinal, prospective collection of data).

The modified sections now read:

[Title] Use of psychological interventions among healthcare workers over the 2-year period following the COVID-19 pandemic: A longitudinal study

[Abstract] Materials and methods: We conducted a longitudinal study on HCWs working in Spain. We used an online survey to collect information on sociodemographic characteristics, depressive symptoms, workplace- and COVID-19-related variables, and the use of psychological support at three time points (2020, 2021, and 2022). Data was available for 296, 294, and 251 respondents, respectively at time points 1, 2, and 3.

[Methods. Study Design and participants] We conducted a longitudinal study including HCWs aged 18 and older working in healthcare facilities in Spain

2. Methods, especially data collection procedure, need to be clarified to avoid confusion. For example, on page 3, line 22, authors said, “wave 1 from April 24th to June 22nd, 2020, n = 2,370 respondents; wave 2 from January 26th 22 to March 8th, 2021, n = 1,807 respondents; and wave 3 form March 23rd to May 23rd 23 , 2022, n = 538)”. Then on page 4, line 5, say said, “Data on the use of psychological support was available for 296, 294, and 251 respondents, respectively, at waves 1, 2, and 3. Two thirds of participants (n = 204, 62%) filled in the three assessment waves, one third (n = 105, 32%) filled in two assessment waves, and 19 (6%) completed one wave only”. I’m not sure which assessment is referred to in the last sentences and who the numbers are related to each other.

Thank you very much for mentioning this section, which indeed requires clarification. As mentioned before, we used what some called an “open cohort”, which means that participants get in and out throughout the study period. This is in addition to the large missing data on the psychological support item, which was placed at the end of the survey and hence left blank by many respondents. We have now added two sentences that we think clarify what the original sample was, what the present sample is, and how many “unique” respondents are at each time point. It now reads:

We conducted a longitudinal study including HCWs aged 18 and older working in healthcare facilities in Spain. We collected self-reported data via an online survey at three different time points (wave 1 from April 24th to June 22nd, 2020, n = 2,370 respondents; wave 2 from January 26th to March 8th, 2021, n = 1,807 respondents; and wave 3 form March 23rd to May 23rd, 2022, n = 538) […] Data on the use of psychological support was available for 296, 294, and 251 respondents, respectively, at waves 1, 2, and 3. These respondents are the sample of the present study. Because recruitment was opened to new participants at each time point, we have a total of 328 unique respondents, of whom 204 (62%) completed the three assessment waves, 105 (32%) completed two, and 19 (6%) completed one.

3. Page 4, line 20, the authors said, “At each follow up assessment, we collected information on use of psychological support in the previous three months (yes/no)”. Was it only collected at the follow up assessment, as it is also mentioned in the baseline (supposing that wave 1 is the baseline)?

Thank you very much for spotting this mistake. We have modified the sentence:

Information on the use of psychological support in the previous three months (yes/no) was collected at all timepoints.

4. What is the association between having depressive symptoms and seeking help/receiving psychological support? It is an important relationship that should be highlighted.

Thank you very much for this. We agree that this is an important association which also supports one of the main implications that we had already included in the Discussion section:

Last, we found only a 10% increase of use of psychological support among people with probable major depressive disorder, compared with people without it. In light of this, we call for improving access to evidence-based scalable interventions at the workplace (see for example (20)).

We also included this in Table 2:

 

Table 2. Association between workplace- and COVID-19-related variables at baseline and use of psychological support at follow-up

 Wave 2 Wave 3

 Odds ratio 95% CI Odds ratio 95% CI

Direct exposure to COVID-19 patients [yes] 2.03 (0.93, 4.41) 1.03 (0.53, 2.03)

Access to protective equipment [yes] 0.81 (0.55, 1.19) 1.10 (0.75, 1.60)

Social stigma for working with COVID-19 patients [yes] 1.20 (0.84, 1.71) 1.13 (0.80, 1.60)

Patient prioritization [yes] 5.59 (2.47, 12.63) 1.45 (0.64, 3.26)

Social support from colleagues [yes] 1.45 (0.88, 2.41) 1.17 (0.75, 1.85)

Probable major depressive disorder (PHQ-9 > 9) [yes] 1.12 (1.06, 1.19) 1.10 (1.04, 1.16)

Note.

PHQ-9 = Patients' Health Questionnaire - 9 items.

And in the Abstract:

Baseline predictors of psychological support were having to make decisions about patients’ prioritisation (OR 5.59, 95% CI 2.47, 12.63) and probable depression (wave 2: OR 1.12, 95% CI 1.06, 1.19; wave 3: OR 1.10, 95% CI 1.04, 1.16).

5. In Table 1, the percentages are not consistent across the table, and it is not clear what they refer to, which makes it confusing to the reader.

Thank you very much. We have included the specific value of the variable to which the frequency and percentage refer to, next to the variable name. We have also added a note to remind the reader that some ordinal variables were dichotomized to facilitate the interpretation of descriptive results (this is detailed in the Methods section). Following Reviewer #1, we also added the percentage of people using psychological support at all timepoints. 

The Table now reads:

 

Table 1. Characteristics of the participants

 Wave 1 Wave 2 Wave 3

 Overall Not received Received Overall Not received Received Overall Not received Received

 (N = 296) (n = 252, 85%) (n = 44, 15%) (N = 294) (n = 243, 83%) (n = 51, 17%) (N = 251) (n = 192, 76%) (n = 59, 14%)

Age in years, Median (IQR) 43 (11) 43 (11) 40 (11) 43 (11) 44 (11) 37 (10) 43 (11) 44 (11) 38 (10)

Gender [female], n (%) 242 (82%) 202 (80%) 40 (91%) 238 (83%) 191 (81%) 47 (92%) 200 (81%) 148 (78%) 52 (91%)

Type of job 

Physician 108 (37%) 92 (37%) 16 (36%) 103 (37%) 86 (37%) 17 (36%) 82 (34%) 64 (35%) 18 (34%)

Nurse 97 (33%) 78 (31%) 19 (43%) 92 (33%) 70 (30%) 22 (47%) 82 (34%) 56 (30%) 26 (49%)

Ancillary workers 36 (12%) 32 (13%) 4 (9.1%) 28 (10%) 24 (10%) 4 (8.5%) 27 (11%) 22 (12%) 5 (9.4%)

Other 53 (18%) 48 (19%) 5 (11%) 54 (19%) 50 (22%) 4 (8.5%) 47 (20%) 43 (23%) 4 (7.5%)

Direct exposure to COVID-19 patients [yes] ¥ 154 (62%) 126 (61%) 28 (68%) 143 (61%) 114 (59%) 29 (74%) 122 (61%) 93 (61%) 29 (62%)

Access to protective equipment [inadequate] ¥ 159 (55%) 133 (54%) 26 (59%) 145 (54%) 115 (52%) 30 (64%) 122 (52%) 97 (53%) 25 (47%)

Social stigma for working with COVID-19 patients [yes] ¥ 91 (31%) 73 (29%) 18 (41%) 85 (31%) 65 (28%) 20 (43%) 71 (30%) 53 (29%) 18 (33%)

Patient prioritization [yes] 48 (25%) 34 (21%) 14 (47%) 40 (23%) 23 (16%) 17 (52%) 38 (25%) 26 (23%) 12 (31%)

Social support from colleagues [yes] ¥ 265 (90%) 226 (90%) 39 (89%) 238 (90%) 194 (89%) 44 (98%) 201 (89%) 155 (88%) 46 (90%)

Probable major depressive disorder (PHQ-9 > 9) [yes] 88 (30%) 69 (28%) 19 (43%) 72 (27%) 51 (24%) 21 (46%) 59 (26%) 41 (23%) 18 (35%)

Note. 

All percentages are valid percentages. PHQ-9 = Patients' Health Questionnaire - 9 items. 

¥ Ordinal variable dichotomized to facilitate the interpretation of the table

---

## [Decision Letter · Decision Letter 1]

2 Oct 2023

Use of psychological interventions among healthcare workers over the 2-year period following the covid-19 pandemic: A longitudinal study

PONE-D-23-19134R1

Dear Dr. Mediavilla,

We’re pleased to inform you that your manuscript has been judged scientifically suitable for publication and will be formally accepted for publication once it meets all outstanding technical requirements.

Kind regards,

Juan Jesús García-Iglesias, Ph.D.

Academic Editor

PLOS ONE

Additional Editor Comments (optional):

Reviewers' comments:

Reviewer's Responses to Questions

**Comments to the Author**

1. If the authors have adequately addressed your comments raised in a previous round of review and you feel that this manuscript is now acceptable for publication, you may indicate that here to bypass the “Comments to the Author” section, enter your conflict of interest statement in the “Confidential to Editor” section, and submit your "Accept" recommendation.

Reviewer #1: All comments have been addressed

Reviewer #2: All comments have been addressed

2. Is the manuscript technically sound, and do the data support the conclusions?

Reviewer #1: Yes

Reviewer #2: Yes

3. Has the statistical analysis been performed appropriately and rigorously? 

Reviewer #1: Yes

Reviewer #2: I Don't Know

4. Have the authors made all data underlying the findings in their manuscript fully available?

Reviewer #1: Yes

Reviewer #2: Yes

5. Is the manuscript presented in an intelligible fashion and written in standard English?

Reviewer #1: Yes

Reviewer #2: Yes

6. Review Comments to the Author

Reviewer #1: (No Response)

Reviewer #2: Dear authors,

Thanks for adressing all my comments. I have no further comments on this manuscript.

7. PLOS authors have the option to publish the peer review history of their article (what does this mean?). If published, this will include your full peer review and any attached files.

Reviewer #1: **Yes: **Fredrick Dermawan Purba, PhD

Reviewer #2: No

---

## [Editor Report · Acceptance letter]

19 Oct 2023

PONE-D-23-19134R1 

Use of psychological interventions among healthcare workers over the 2-year period following the COVID-19 pandemic: A longitudinal study 

Dear Dr. Mediavilla:

I'm pleased to inform you that your manuscript has been deemed suitable for publication in PLOS ONE. Congratulations! Your manuscript is now with our production department. 

Kind regards, 

on behalf of

Dr. Juan Jesús García-Iglesias 

Academic Editor

PLOS ONE